# Enolase Is Implicated in the Emergence of Gonococcal Tolerance to Ceftriaxone

**DOI:** 10.3390/antibiotics12030534

**Published:** 2023-03-07

**Authors:** Sheeba Santhini Manoharan-Basil, Margaux Balduck, Saïd Abdellati, Zina Gestels, Tessa de Block, Chris Kenyon

**Affiliations:** 1HIV/STI Unit, Department of Clinical Sciences, Institute of Tropical Medicine Antwerp, 2000 Antwerp, Belgium; 2Clinical Reference Laboratory, Department of Clinical Sciences, Institute of Tropical Medicine, 2000 Antwerp, Belgium; 3Department of Medicine, University of Cape Town, Cape Town 7700, South Africa

**Keywords:** tolerance, enolase, ceftriaxone, Neisseria gonorrhoeae

## Abstract

Antibiotic tolerance is associated with antibiotic treatment failure, and molecular mechanisms underlying tolerance are poorly understood. We recently succeeded in inducing tolerance to ceftriaxone (CRO) in an *N. gonorrhoeae* reference isolate. In a prior in vitro study, six biological replicates of WHO P strains were exposed to CRO (10× the MIC) followed by overnight growth, and tolerance was assessed using a modified Tolerance Disc (T.D.) test. In the current study, we characterized the mutation profile of these CRO-tolerant phenotypes. The whole genome was sequenced from isolates from different replicates and time points. We identified mutations in four genes that may contribute to ceftriaxone tolerance in *N. gonorrhoeae*, including a mutation in the enolase (*eno*) gene that arose independently in three lineages.

## 1. Introduction

Gonorrhoea, a bacterial sexually transmitted infection (STI), is caused by *Neisseria gonorrhoeae* [1,2,3]. It is the second-most-common bacterial STI worldwide [4]. *N. gonorrhoeae* has developed resistance to multiple classes of antibiotics, including ceftriaxone (CRO), the recommended treatment for gonorrhoea [5,6,7].

Antibiotic tolerance or persistence is defined as the ability of a bacterial subpopulation to survive high antibiotic concentrations to which the bacteria are fully susceptible without an increase in minimum inhibitory concentration (MIC) [8,9]. Antibiotic tolerance can contribute to treatment failure and the emergence of antibiotic resistance [10,11,12,13]. Tolerance-related treatment failure can occur when a subpopulation of bacteria in a dormant state survives the lethal effect of the antibiotic [8,9]. Tolerance to antibiotics has been shown to play a role in the clinical persistence of infections caused by *Escherichia coli*, *Pseudomonas aeruginosa* and *Mycobacterium tuberculosis* [13,14,15]. Tolerance has also been shown to emerge prior to, and facilitate the emergence of, antimicrobial resistance (AMR) in *E. coli* [16]. Thus, tolerance mutations may pave the way for a rapid subsequent evolution of AMR.

We recently succeeded in inducing tolerance to ceftriaxone (CRO) via intermittent cyclic CRO exposure in *N. gonorrhoeae* reference isolates (WHO P) [17]. In the current study, using whole-genome sequencing (WGS), we characterized the mutation profile of these CRO-tolerant phenotypes.

## 2. Material and Methods

### 2.1. Phenotypes

We recently induced tolerance to ceftriaxone (CRO) in *N. gonorrhoeae* reference isolates (WHO P) [17]. A highly susceptible CRO reference strain, WHO P with a CRO MIC of 0.004 µg/mL, was used [18]. Briefly, a direct colony suspension method was used, wherein WHO P reference colonies from overnight cultures were suspended in Gonococcal (GC) broth (containing 15 g/L bacto protease peptone, 1 g/L soluble starch, 4 g/L K_2_HPO_4_ (174.18 g/mol), 1 g/L KH_2_PO_4_ (136.08 g/mol), 5 g/L NaCl (58.44 g/mol), supplemented with 1% BD BBLTM IsoVitaleX,) adjusted to a turbidity of 0.5–1.0 McFarland (McF) and exposed to a high concentration (conc.) of CRO (10× the MIC (0.04 µg/mL)) for a fixed duration (3 h) in a cyclic manner. After each cycle, overnight cultures were suspended in (GC.) broth (50 µL in 5 mL), containing 0.04 µg/mL of final conc. of CRO antibiotic (Merck Life Science, Darmstadt, Germany). After this, antibiotics were removed by washing the cultures twice with 10 min centrifugation (1400× *g*) and resuspending the pellets (1 mL) in fresh GC. broth (5 mL) and incubated overnight (21 h) at 36 °C in a 6.0% CO_2_ incubator. The experiment was carried out in 15 mL falcon tubes placed on a roller-mixer (RM 5, CAT, Staufen, Germany). Following each exposure cycle, the CRO MIC was determined using a CRO gradient E-test ranging from 0.016 to 256 µg/mL (BioMérieux, France). The samples were stored at −80 °C in skim milk containing 30% glycerol. The cyclic CRO interval exposure experiment was carried out for six biological replicates for 7 consecutive days. Six and two populations were tested with and without the antibiotic, respectively.

To detect the presence of tolerant phenotypes, the CRO-exposed population was analyzed using a modified T.D. test protocol (Figure 1; [19]). Briefly, the samples, after the CRO exposure cycle, were inoculated on BD^TM^ Columbia Agar with 5% Sheep Blood and incubated at 36 °C overnight (18–24 h) in a 6.0% CO_2_ incubator. This was followed by a T.D. test wherein 0.008 µg of CRO antibiotic was added to a 6 mm Whatmann^®^ antibiotic assay disc. After ~18 h of incubation, the CRO discs were replaced by nutrient discs, i.e., discs containing G.C. broth, and incubated overnight. Additionally, 10 µL of G.C. medium was added to the same nutrient disc and incubated for an additional night as *N. gonorrhoeae*-tolerant colonies emerged 48 h after adding the nutrient disc. The tolerant colonies were inoculated on fresh blood agar plates and stored in 30% glycerol skim milk at −80 °C.

### 2.2. Whole-Genome Sequencing (WGS) and SNP Analysis

In this study, 16 tolerant isolates from 6 populations evolved with CRO antibiotic, and two control populations evolved without antibiotic were subjected to Whole-Genome Sequencing (WGS; Table 1). Sequencing of the tolerant isolates was outsourced to Eurofins, where total DNA was isolated from tolerant (n = 14) and control strains (n = 2). Library preparation was carried out using a Stranded TruSeq DNA library preparation kit. The sequencing was performed on NextSeq6000, v2, 2 × 150 bp (Illumina Inc., San Diego, CA, USA), followed by analysis of the raw reads described in [20]. For the WGS analysis, initial quality control (QC) of the raw reads was carried out using FASTQC [19]. The raw reads were trimmed using trimmomatic (v0.39) (Phred score ≥ 20 and length of the bases ≥32 bases). The processed raw reads were de novo assembled using Shovill (v1.0.4) [21], which uses SPAdes (v3.14.0) using the following parameters: trim—depth 150—opts—isolate. The quality of the de novo assembled contigs was evaluated; using Quast (v5.0.2) [22] and annotated using Prokka (v1.14.6) [23]. The quality-controlled reads were mapped to control draft genome using BWA MEM, and single-nucleotide polymorphisms (SNPs) were determined using freebayes implemented in Snippy (v4.6.0) with default parameters (10× minimum read coverage and 90% read concordance at the variant locus) [24,25,26]. The raw reads generated are deposited at https://www.ncbi.nlm.nih.gov/bioproject/PRJNA924144 (accessed on 16 January 2023).

### 2.3. Genetic Characterization of Eno and tatC Genes Associated with Ceftriaxone Tolerance in WHO-P and Global Neisseria spp. Collection

The putative SNPs identified in the relevant genes associated with ceftriaxone tolerance were further examined in the global collection comprising the genomes of *N. gonorrhoeae* (n = 17,871), including WHO-P and commensal *Neisseria* spp. (n = 1136), whose provenance and metadata are described elsewhere [21].

## 3. Results

We summarize the emergent mutations in Figure 2 and Table 2. WGS analysis revealed mutations in *tatC* (twin arginine translocase), *edd* (Phosphogluconate dehydratase), *eno* (enolase) and C7S06_RS11330 genes (uncharacterized protein). 

Substitutions in twin-arginine translocase (Met63Ile) and Phosphogluconate dehydratase (Ala510Val) were observed at timepoint 7 in lineage 1 (Isolate ID 16.3–2.3). Three different substitutions were observed at time point 7 in lineages 3 (Gly135Asp; Isolate ID 16.5–2.3), 4 (Gly138Asp; Isolate ID 16.6–2.3) and 5 (Gly30Val; Isolate ID 16.7–2.3) for the enolase protein at timepoint 7. The substitution in C7S06_RS11330 occurred at timepoint 1 in lineage 6 (Isolate ID 10.8; Figure 2 and Table 2).

None of the above mutations were observed in the global collection of 17,871 *N. gonorrhoeae* isolates. 

Of note, no tolerant colonies were observed in lineage 1 and all tolerant WHO P colonies were found to have a CRO MIC ≤ 0.008 µg/mL, comparable to the MIC of the control samples. No increase in CRO MIC was observed for either the tolerant or the control isolates.

## 4. Discussion

Previously, we reported that tolerance to CRO can be induced in CRO-susceptible (<0.004 µg/mL) *N. gonorrhoeae* WHO P reference isolate [17]. In the current analysis, we describe, for the first time, the mutations associated with emergence of CRO tolerance in *N. gonorrhoeae.*

Enolase, a glycolytic enzyme, which catalyzes the conversion of 2-phospho-D-glycerate to phosphoenolpyruvate, is involved in carbon metabolism [22]. It is also a component of the RNA degradosome, which is involved in RNA processing and gene regulation [23,24]. In *Pseudomonas aeruginosa*, enolase influences tolerance to oxidative stress by affecting the production of *ahpB* and *ahpC* in an OxyR-independent manner [25]. In another study, oxidative stress response genes *gor* and *ahpC* were found to play a role in antibiotic tolerance of *Streptococcus mutans* biofilms [26]. In CRO-tolerant *N. gonorrhoeae,* a mutation in the enolase gene arose independently in three lineages at time point 7, implicating enolase in CRO-tolerance. 

Mutations in *tatC* and *edd* genes were identified at timepoint 7 in lineage 1. Mutations in *tatC* and other genes, *gltI, hlpA*, *ruvC*, *ddlB* and *ydfI,* were found to result in tolerance to tosufloxacin in *E. coli* [27]. TatC, the primary substrate receptor, is part of the twin-arginine translocation (Tat) system needed to transport folded proteins across biological membranes [28]. In *Zymomonas mobilis* the efficient export of NADP-containing glucose-fructose oxidoreductase to the periplasm depends on an intact twin-arginine motif [29] and, thus, although the mutations in *tatC* were identified only at one time point, these mutations may be implicated in tolerance in *N. gonorrhoeae.*

We were unable to perform the complementation experiments necessary to prove that the mutations we identified played a role in generating tolerance. Other limitations of our study include the small sample sizes and the fact that we only used one antimicrobial to assess the pathway to tolerance. Future studies could remedy these shortcomings as well as assess if tolerance plays a meaningful role in gonococcal treatment failure and the emergence of AMR, as has been shown for other bacterial species [13].

Our global phylogenetic analysis revealed that none of the isolates had the above mutations, which suggests that these mutations might have a high fitness cost and, therefore, not be seen in the natural population. This also raises the possibility that these putative-tolerance-associated mutations are transient. A number of studies have found that transient mutations can act as stepping stones to antibiotic resistance in *N. gonorrhoeae* and other bacteria [30,31]. Several, in vitro antibiotic exposure experiments have shown that the development of AMR is always preceded by the emergence of tolerance, in which several partial resistance mutations can occur [11,14,16,32,33]. Our findings raise the possibility that transient mutations may, likewise, emerge in response to exposure to certain antimicrobials. These may, in turn, facilitate the emergence of AMR, as has been shown in *E. coli* [16]. Interestingly, isolates from lineages 3, 4 and 5 with a mutation in the *eno* gene did not have mutations in *tatC* and *edd* genes and vice versa, suggesting that there may be multiple pathways involved in CRO tolerance in *N. gonorrhoeae*. 

## Figures and Tables

**Figure 1 antibiotics-12-00534-f001:**
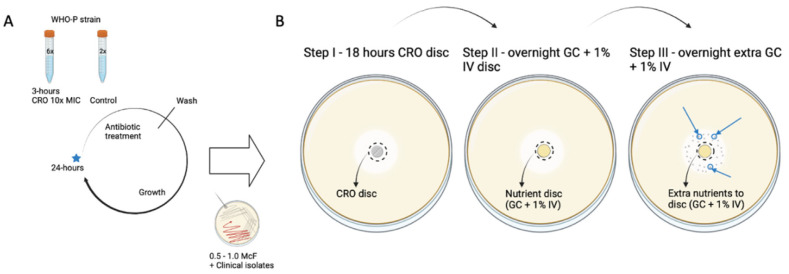
Induction of tolerance to CRO in WHO P *N. gonorrhoeae* reference strain and tolerance detection via TD test. (**A**) Cyclic 3 h 10× MIC ceftriaxone interval exposure (n = 6) with two control samples not exposed to CRO. (**B**) TD test after every 24 h exposure cycle. Blue star indicates the point at which cyclic exposure samples were inoculated, tested for MIC, and stored. Blue arrow indicates emergent tolerant colonies at Step III. Figure is reproduced with permission from [17].

**Figure 2 antibiotics-12-00534-f002:**
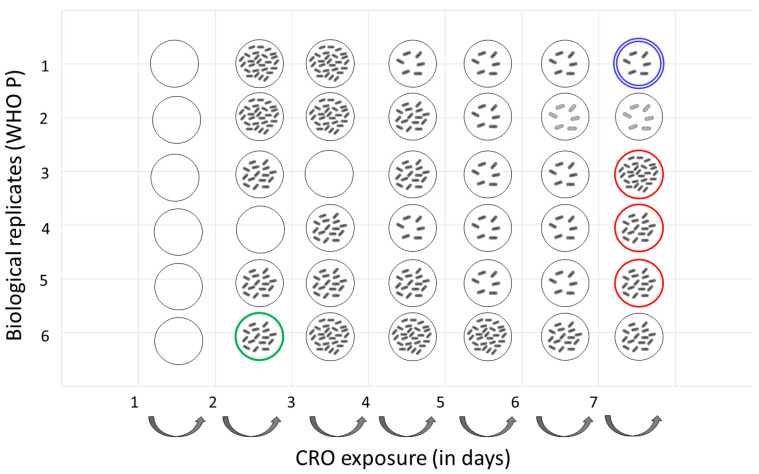
Ceftriaxone tolerance evolution of *N. gonorrhoeae* WHO P strain (n = 6) after 3 h 10× MIC cyclic exposure to CRO. Tolerance was determined using TD test. TD test results were categorized as follows: low tolerance (1–10 colonies), medium tolerance (10–20 colonies) and high tolerance (>20 colonies). Double halo (growth around inhibition zone) is depicted in light gray (Time point 6 and 7 of lineage 1). Red and green circles denote isolates with mutation in *eno* and C7S06_RS11330 genes, respectively. Blue circle denotes isolates with mutations in *tatC* and *edd* genes. Empty circle denotes no tolerant isolates.

**Table 1 antibiotics-12-00534-t001:** Evolution of ceftriaxone tolerance over time (cycles 1, 2.7). The isolates that were subjected to whole-genome sequencing are colored in grey and the numbers denote the isolate identification (id).

		Exposure Cycles						
	Biological Replicates	1	2	3	4	5	6	7
Control isolates	1	/	/	12.1	/	/	/	16.1
	2	/	/	/	/	/	/	/
Tolerant isolates	1	/	11.3–2.3	/	/	/	/	16.3–2.3
	2	/	11.4–2.3	/	/	/	/	16.4–2.3
	3	/	11.5–2.3	/	/	/	/	16.5–2.3
	4	/	/	12.6.3	/	/	/	16.6–2.3
	5	/	11.7–2.3	/	/	/	/	16.7–2.3
	6	10.8	11.8–2.3	12.8.3	/	/	/	16.8–2.3

**Table 2 antibiotics-12-00534-t002:** List of variants identified in the study.

Protein Product	Gene	Lineages	Cycle	Isolate ID	CDS Change	Amino acid Change
Helix-hairpin-helix domain-containing protein	C7S06_RS11330	L6	1	10.8	205A > C	Ile69Leu
Twin-arginine translocase subunit TatC	*tatC*	L1	7	16.3–2.3	189G > A	Met63Ile
Phosphogluconate dehydratase	*edd*	L1	7	16.3–2.3	1529C > T	Ala510Val
Phoshopyruvate hydratase	*eno*	L3	7	16.5–2.3	404G > A	Gly135Asp
		L4	7	16.6–2.3	413G > A	Gly138Asp
		L5	7	16.7–2.3	89G > T	Gly30Val

## Data Availability

The raw reads generated are deposited at https://www.ncbi.nlm.nih.gov/bioproject/PRJNA924144 (accessed on 16 January 2023).

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
