# Peer review of "Enolase Is Implicated in the Emergence of Gonococcal Tolerance to Ceftriaxone"

_antibiotics, 2023, doi:10.3390/antibiotics12030534_

Round 1

Reviewer 1 Report

In this study the authors sequence the genomes of in vitro evolved Ngo lineages exposed to CRO. They identify CRO tolerant lineages and identify 4 putative mutations that may underly the phenotype. This is an interesting study that will be of interest to others in their field.  

Line 11: It may be helpful to specify that tolerance was induced through In vitro evolution.

Line 12: Add “In a prior study” to begin the sentence starting on this line.

Line 15: Different lineages here suggests different clinical strains. However I believe this means different experimental replicates. Please clarify.

Line 52: Why was 0.04 ug/ml chosen as the concentration to evolve the WHO P isolate?

Line 53: Define (conc.). This abbreviation has not been previously used.

Line 59: It is confusing how many replicates were evolved in each condition. Please rewrite this sentence for clarity.

Line 81: denovo is two words. Please modify.

Table 1: Please add a table legend clarifying what the shading and numbers mean in each cell.

Line 143: Why were complementation experiments not completed?

Line 150: It may be worth discussing the possibility that these mutations have a high fitness cost and that is why they are not seen in natural populations.

General comment: Why was the WHO P strain selected to evolve tolerance? Please provide a rationale if there is one.

Author Response

Dear Reviewer,  

We thank you for the opportunity to improve and resubmit our manuscript. We thank you for your insightful comments and feedback on the manuscript. The questions have been carefully considered and answered to improve the manuscript to the best of our ability. Changes to the manuscript are available as tracked changes. The responses to all comments are provided in the attached document.

Reviewer 2 Report

The topic of the article is relevant and interesting enough. However, the presented results are preliminary in nature and there is a lack of reliable information supporting the main hypothesis. There is no statistical analysis of the data, which reduces the reliability of the results obtained.

Author Response

Dear Reviewer,  

We thank you for your  comments and feedback on the manuscript. We have responded to the question to the best of our ability. 

Response 1: 

Though the results are preliminary, mutation in enolase gene in CRO-tolerant N. gonorrhoeae, arose independently in three lineages at time point 7, and verified by whole genome sequencing, implicating enolase in CRO-tolerance. The small sample size has been mentioned as a limitation in lines 163-165.

Reviewer 3 Report

The paper presents very interesting and important results on  antibiotic tolerance of N. gonorrhoeae. The previously  unknown mutation has been  detected, which could appear to be very important for the drug resistance.

Minor remarks

(1) In my opinion, it should be added to the results that no mutations associated with resistance to ceftriaxone and other drugs of the beta-lactam series were found in tolerant strains (according to the WGS results).

(2) Did the authors detemine MIC to ceftriaxone and other antibiotics? It would be very interesting to see whether any  changes in MIСсro anf MICpen are observed in tolerant strains, even very small changes.

Author Response

Reviewer 3

The paper presents very interesting and important results on  antibiotic tolerance of N. gonorrhoeae. The previously  unknown mutation has been  detected, which could appear to be very important for the drug resistance.

Dear Reviewer,  

We thank you for the opportunity to improve and resubmit our manuscript. We thank you for your insightful comments and feedback on the manuscript. The questions have been carefully considered and answered to improve the manuscript to the best of our ability. Changes to the manuscript are available as tracked changes. The responses to all comments are provided below in red colour font.

Minor remarks

  • In my opinion, it should be added to the results that no mutations associated with resistance to ceftriaxone and other drugs of the beta-lactam series were found in tolerant strains (according to the WGS results).

Response 1: Thank you for this suggestion. Since there were no increase in MICs of the tolerant colonies, the mutations associated with ceftriaxone resistance and the beta-lactams were not analysed in WGS and therefore not mentioned.

  • Did the authors determine MIC to ceftriaxone and other antibiotics? It would be very interesting to see whether any  changes in MIСсro and MICpen are observed in tolerant strains, even very small changes.

Response 2: We determined MIC to ceftriaxone but not to other antibiotics. This was mentioned in the previous study but we failed to mentioned in the current study. We apologise for this and have added the following to lines 64 to 66 and 123-126, “Following each exposure cycle, the CRO MIC was determined using a CRO gradient E-test ranging from 0.016 to 256 µg/ml (BioMérieux, France)”.

Of note, no tolerant colonies were observed in lineage 1 and all tolerant WHO P colonies were found to have a CRO MIC ≤ 0.008 µg/mL, comparable to the MIC of the control samples. No increase in CRO MIC was observed for either the tolerant or the control isolates.

Reviewer 4 Report

In the brief report entitled “Enolase is implicated in the emergence of gonococcal tolerance to 

ceftriaxone”, the authors Sheeba S. Manoharan-Basil *, Margaux Balduck, Saïd Abdellati, Zina Gestels, Tessa de Block, Chris Kenyon study the emergence of antibiotic tolerance as a consequence of cyclic exposure to high concentrations of ceftriaxone for a limited amount of time, followed by incubations with lower levels or without antibiotic. Through these cycles, the authors isolate lineages that can withstand growth in the presence of ceftriaxone using the tolerance disk tests. The authors later sequence these lineages and identify 4 genes that display mutations compared to a collection of genomes from N. gonorrhoeae. Interestingly, the authors identify 3 lineages that display a mutation in the gene encoding enolase, as well as other lineages. From their analysis none of these mutations has been observed in clinical samples, however mutations in the identified genes have been observed in other species such as E. coli and Zymomonas mobilis

Overall, the manuscript is well-written and concise. The English language is intelligible and coherent. It does not contain grammar or spelling errors. The research problem is well-presented, the workflow coherent and sound, and the analysis is complete. The methods support the research question established and are well-described, except at specific points mentioned below. 

Minor points:

Introduction:

  • L. 27: I believe the current phrase “tolerance or antibiotic persistence” should be “Antibiotic tolerance or persistence”.

Materials and Methods:

  • Overall, the cyclic exposure of samples to CRO remained unclear at points:

  • L. 48: The exposure is performed in liquid or on agar?

  • L. 48 and 52: state the same concentration of MIC. It is unclear whether that concentration corresponds to 10xMIC or to 1xMIC.

  • If the concentrations are the same, then the samples are exposed to 10xMIC for (3h+o/n). Is this how the protocol works? 

  • Potentially providing a concise diagram of the cyclic exposure could help the reader:

  • L. 55: the 1:5 ratio relates to the resuspension of cells or to the dilution of the GCB? Eg, 1 vol of cells (5 mL pellets) to 5 volumes of 1xGCB (25 mL) OR pellets resuspended in diluted GCB (0.167x).

  • L. 59: delete “/cycles”. 

  • L. 59: to be sure, in 7 days, the population went through 3 CRO cyclic exposures, as it is (3h with 10x MIC CRO+ o/n with 1xMIC CRO + o/n without CRO) / cycle, correct?

  • L. 60: “were tested” instead of “evolved”

  • L. 61-64: once the tolerant phenotypes are isolated, they are plated on agar and incubated. It is unclear for how long the isolates are incubated prior to applying the tolerance disks. Please precise the incubation duration.

  • L. 64-69: For clarification, between the inoculation of samples (l. 63) and the observation of tolerant colonies (l. 69) 72h have passed, is this correct? 

Results:

  • The diagram in Fig 1 was not clear to me. On the x-axis, CRO exposure cycles correspond to days or to full cycles of exposure? On the x-axis why are the numbers on the axis in between the observations?

  • The icons in the image correspond to bacteria? Is this a representation of the cell density at each exposure cycle? At what point of the exposure cycle? Why is “1,” empty? Why are 2,4 and 3,3 empty?

  • The light gray was not obvious on the image. I see light gray “bacteria” on 2,6 and 2,7, but Table 2 states that only lineage 1 had mutations. 

  • In the results sections, it should be explicitly stated that lineage L2 did not display any tolerant colonies.

  • Have the authors assessed the resistance profiles of the tolerant colonies? What is the phenotype of these isolates when exposed to CRO MIC plates or to CRO disk diffusion assay?

Discussion:

  • While the importance of antibiotic tolerance is described in the introduction, it might be worth emphasizing again in the discussion the overall impact of tolerance in the global antimicrobial resistance crisis. 

I believe this manuscript is of great interest to the reader. The antimicrobial resistance crisis in Neisseria gonorrhoeae continues to grow and is becoming ever more dramatic. It is necessary for scientists to have all the tools necessary to help combat this crisis. This manuscript adds to the body of knowledge by performing experimental evolution to assess the capacity of N. gonorrhoeae to adapt to high levels of antimicrobial concentrations. 

Except for the few clarifications requested in the above review, I am confident that this manuscript is of great quality.

Author Response

(The authors gave the same response as above.)

Round 2

Reviewer 2 Report

Dear Authors,

I accept the changes and corrections made. However, I still believe that the small sample size and the lack of statistical analysis make the results somewhat debatable.